# Changes in Antibiotic Prescribing Patterns in Danish General Practice during the COVID-19 Pandemic: A Register-Based Study

**DOI:** 10.3390/antibiotics11111615

**Published:** 2022-11-13

**Authors:** Camilla Rask Nymand, Janus Laust Thomsen, Malene Plejdrup Hansen

**Affiliations:** Center for General Practice, Aalborg University, 9220 Aalborg, Denmark

**Keywords:** COVID-19, general practice, antibiotic agents, primary health care, antimicrobial agents

## Abstract

The World Health Organization expressed concern that antimicrobial resistance would increase during the COVID-19 pandemic due to the excessive use of antibiotics. This study aimed to explore if antibiotic prescribing patterns in general practices located in the North Denmark Region changed during the COVID-19 pandemic. The study was conducted as a registry-based study. Data was collected for every antibiotic prescription issued in general practices located in the North Denmark Region during the first year of the pandemic (1 February 2020 to 31 January 2021) and the year prior to the pandemic (1 February 2019 to 31 January 2020). Data were compared regarding antibiotic agents and the type of consultation linked to each antibiotic prescription. Results showed that antibiotic prescriptions decreased by 18.5% during the first pandemic year. The use of macrolides and lincosamides, along with combinations of penicillins and beta-lactamase -sensitive penicillins, was reduced the most. Face-to-face consultations related to an antibiotic prescription decreased by 28.5%, while the use of video consultations increased markedly. In Denmark, COVID-19 restrictions have contributed to both a lower consumption of antibiotics and a change in prescription patterns in general practice. Probably some of the COVID-19 -preventing initiatives could be of importance moving forward in the fight against antimicrobial resistance.

## 1. Introduction

The discovery of antibiotics in 1928 became a revolution in modern medicine [1]. Today antibiotics are an integrated part of daily medical practice in both hospitals as well as in the primary health care sector. However, if we do not use antibiotics with caution, we risk returning to a pre-antibiotic era, where simple infections can result in increased morbidity and even death [2,3]. Any antibiotic use contributes to the increasing problem of antimicrobial resistance (AMR). According to the World Health Organization (WHO), AMR is considered one of the top 10 global public health threats facing humanity [4]. In Denmark, a surveillance system called DANMAP (The Danish Integrated Antimicrobial Resistance Monitoring and Research Programme) has been developed [5]. DANMAP was established in 1995 by the Danish Ministry of Food, Agriculture and Fisheries and the Danish Ministry of Health to study associations between antibiotic consumption and antimicrobial resistance and to monitor the consumption of antimicrobial agents for food, animals and humans across the Danish primary and secondary healthcare [5]. However, the DANMAP reports mainly report on total data from the Danish primary healthcare system. Consequently, it is not possible to identify antibiotic scripts issued in different settings, such as for example general practice, private ear–nose–throat specialists and gynaecologists or dentist clinics.

For more than a decade, antibiotic use in the Danish primary healthcare sector has steadily decreased from 17.05 DID (2011) to 12.83 DID (2020) [6,7]. In addition, during the first year of the Coronavirus Disease 2019 (COVID-19) pandemic, an overall reduction of antibiotic use in both the Danish primary and secondary healthcare sectors was observed [7]. Danish primary health care comprises several types of medical specialists here, including gynaecologists, otolaryngologists, dermatologists, dentists and general practitioners. Importantly, the majority of antibiotics used in the primary healthcare sector in Denmark are issued in the general practice setting [8].

The reduction in overall antibiotic use contradicts WHO’s anticipations that AMR would rise during the pandemic due to inappropriate use of antibiotics [9]. The reduced antibiotic consumption might be explained by the focused attempts to minimise severe acute respiratory syndrome coronavirus 2 (SARS-CoV-2) transmission in society and in medical settings. These initiatives may have led to reduced viral as well as bacterial transmission resulting in a reduced number of infections that required antibiotic treatment [10]. However, to our knowledge, no previous Danish study has scrutinised exclusively antibiotic consumption in general practice. These included the introduction of video consults [11,12], face shields and avoidance of seeing patients with symptoms of COVID-19 infection in general practices. Similar initiatives were commenced in other countries, where several studies have found a reduction in antibiotic use in the general practice setting. This includes studies performed in the Netherlands, UK, Australia and New Zealand [13,14,15,16,17,18].

This study aimed to describe if the antibiotic prescribing patterns in general practices located in the North Denmark Region changed during the COVID-19 pandemic compared to one year prior.

## 2. Results

### 2.1. Antibiotic Prescribing Rates during the COVID-19 Pandemic

A total reduction of 35,637 antibiotic prescriptions was observed for the pandemic period compared to the pre-pandemic period (*p*-value < 0.0001). This finding corresponds to an overall decrease of 18.5% (Table 1). Table 1 provides an overview of the number of antibiotic prescriptions issued each month.

Upon the Danish national lockdown, initiated on 11 March 2020, a considerable decrease was observed for the upcoming month in the antibiotic prescribing rates from 16,918 in March 2020 to 12,067 in April 2020. A considerable decrease in monthly prescribing rates was seen during the COVID-19 pandemic compared to the pre-pandemic period. From April 2020 onward, the monthly reduction varied from 13% to 35% (Figure 1).

### 2.2. Antibiotic Prescribing Patterns Based on ATC-Groups

All types of antibiotics, except nitrofurantoin, were prescribed to a lesser extent during the pandemic compared to the previous year (Figure 2). A 45.4% decrease in the use of macrolides and lincosamides was seen during the pandemic. In addition, combinations of penicillins (31.6%), beta-lactamase sensitive penicillins (29.1%) and sulphonamides/trimethoprim (19.4%) were used to a lesser extent during the pandemic compared to the previous year. Only a minor reduction in the use of beta-lactamase -resistant penicillins was observed (4.2%). The use of nitrofurantoin increased by 20.7% (n = 1028 prescriptions).

The distribution among the various types of antibiotics remained the same during both observation periods despite an overall reduction in the number of antibiotic prescriptions issued during the two periods (*p*-value: 0.067).

### 2.3. Consultation Types Related to Antibiotic Prescribing Patterns

Figure 3 shows the various consultation types in general practice linked to an antibiotic prescription before and during the COVID-19 pandemic. Overall, 31,619 fewer consultations linked to an antibiotic prescription were found for the pandemic period compared to the pre-pandemic period (−19.6%). Face-to-face consultations were the preferred type of consultation when issuing an antibiotic prescription, both before and during the pandemic. However, the number of face-to-face consultations decreased by 28.5% during the pandemic. Telephone consultations resulting in an antibiotic prescription increased by 14.6%, while video consults related to an antibiotic prescription increased by more than 1000-fold during the pandemic.

The total number of missing data regarding consultation types linked to an antibiotic prescription was 31,430 for the pre-pandemic period and 27,412 for the pandemic period.

## 3. Discussion

### 3.1. Principal Findings

An 18.5% reduction in antibiotic prescriptions issued in general practices located in the North Denmark Region was observed during the COVID-19 pandemic. The most prominent reduction was seen for the use of macrolides and lincosamides (J01F), while a minor increase in the use of nitrofurantoin (J01XE) was observed. The large reduction in the use of macrolides and lincosamides can perhaps be explained by a change in the Danish venereology guidelines in 2019 for treating *Chlamydia trachomatis* infection with tetracyclines instead of macrolides. Furthermore, restrictions on issuing azithromycin were established in the Danish primary healthcare sector during the pandemic [7,19]. The minor increase in nitrofurantoin prescriptions can possibly be explained by COVID-19 restrictions having no effect on the incidence of urinary tract infections (UTI), for which nitrofurantoin is solely used in Denmark [20].

Combinations of penicillins (J01CR) and beta-lactamase sensitive penicillins (J01CE) also decreased considerably during the pandemic, which must be directly linked to the COVID-19 initiatives introduced to minimise acute respiratory tract infections, where both antibiotic agents are often prescribed in Danish general practice. A finding of great potential moving forward.

The consultation types related to antibiotic prescribing changed significantly during the pandemic. Face-to-face consultations decreased—yet remained the most frequent consultation type related to an antibiotic prescription. The use of video consultations increased markedly during the pandemic.

### 3.2. Findings in Relation to Other Studies

The findings of this study are in line with studies from other countries, including studies from the Netherlands [13,14,15,16,17,18]. The Netherlands has a similar antibiotic prescribing pattern, although the total antibiotic consumption is lower than in Denmark [21]. We found an almost 30% reduction in the use of phenoxymethylpenicillin during the pandemic. This reduction is probably caused by the reduced number of patients attending general practice with symptoms of acute respiratory tract infections. This finding is in accordance with a Dutch study that reported a decrease in the use of antibiotics for acute respiratory tract infections [14]. The Dutch study suggested, in addition, that this reduction was probably due to social distancing [14].

This study demonstrates an 18.5% reduction in the use of antibiotics in general practices in the North Denmark Region during the COVID-19 pandemic. Interestingly, the Danish national surveillance system DANMAP reports a reduction of only 6.8% in the overall antibiotic consumption in the Danish primary healthcare sector in 2020 compared to 2019 [7]. This difference might be caused by an increase in antibiotic consumption in other primary healthcare settings, such as for example dental clinics or gynaecologists. Interestingly, studies from England and Scotland have shown an upward trend in antibiotic prescribing by dentists during the pandemic [22,23]. In addition, the DANMAP report places the North Denmark Region as the median of the five Danish regions [7] concerning antibiotic consumption. Consequently, a larger antibiotic use in the other regions may have contributed to a smaller reduction on a national level.

The reduced use of antibiotics in general practice during the COVID-19 pandemic can be explained by both the restrictions introduced in society and on a personal level, especially including social distancing to prevent infective transmission. Other initiatives, such as the use of hand sanitisers and face masks, may also have contributed to diminishing infections, although it is questionable if the effect of face masks is sufficient in preventing virus transmission [24,25]. The large reduction in antibiotic prescribing rates does raise the question of whether patients were underdiagnosed or undertreated during the pandemic. DANMAP reported in their annual seminar (November 2021) how no change in invasive infections was found during the first pandemic year compared to previous years—apart from infections caused by *Haemophilus influenza* and *Streptococcus pneumoniae*, where a reduction in invasive infections was reported dismissing the matter in question [7].

A marked change in the type of consultation linked to the prescribing of antibiotics was seen. The most obvious explanation is that most patients with symptoms and signs of acute infection were booked for a video consultation instead of a face-to-face consultation. Furthermore, telephone consultations linked to an antibiotic prescription increased during the pandemic, consistent with the total number of telephone consultations increasing significantly in the entire primary healthcare sector [7]. In addition, in New Zealand [26], the UK [27], and Australia [28], fewer face-to-face consultations and more remote consultations were observed. In New Zealand, this change was primarily doctor driven to protect the doctors’ from being infected by COVID-19 and thereby securing continued access to health care [26].

A 7% increase in the total number of consultations in Danish general practice in 2020 compared to 2019 has been reported [7]. Despite this increase in the overall number of consultations, this study found a 19.6% reduction in consultations related to antibiotic prescriptions. This finding accentuates GPs prescribing fewer antibiotics during the COVID-19 pandemic.

### 3.3. Strengths and Limitations

This study included information exclusively from general practices in the North Denmark Region issuing one or more antibiotic prescriptions within the study period. When interpreting the results of this study, some limitations must be kept in mind. The study measured antibiotic consumption by the number of prescriptions issued by a GP. This method does not accommodate the amount or dose of antibiotic used, which is an important element when considering the increasing problems with resistant bacteria. The use of Defined Daily Doses per 1000 inhabitants per day (DID) could have been implemented to accommodate this limitation. However, an assumption can be made that the same guidelines for treating infections in Danish general practice have been followed during the two periods, and, thereby, prescriptions for the same infectious diseases are alike. Due to the data source (the North Denmark Region prescription database), it is not possible to obtain information about whether patients redeemed their prescriptions at the pharmacies. Nor whether the patients were compliant in administering the antibiotics as prescribed by the GP. Finally, information about consultation type was not available for all antibiotic prescriptions resulting in a rather large proportion of missing data for this outcome.

### 3.4. Implications

COVID-19 restrictions have contributed to lower consumption of antibiotic agents by reducing viral and bacterial transmission in society. Antibiotic consumption can potentially remain at this low level if restrictions and behaviours are continued beyond the pandemic. However, several restrictions are hard to implement permanently, such as social distancing without a reduction in quality of life. Contrariwise, other restrictions such as frequent hand sanitising, applying face masks when having an acute respiratory tract infection, and staying or working from home when sick could be implementable in the future in order to prevent viral and/or bacterial transmission. Furthermore, initiatives in general practices could be implemented beyond the pandemic, such as, for example, the use of video consultations for patients with symptoms of acute infection and thereby continue to minimise bacterial and viral transmission to both medical professionals and other patients. By January 2022, video consultations will be established as a permanent solution and option for consultations in Danish general practice [29,30].

Lastly, it would be very relevant to also explore how medical specialists other than GPs in the Danish primary healthcare sector have prescribed antibiotics during the COVID-19 pandemic. Not only to estimate the exact numbers for the different settings but also to clarify the exact contribution of Danish general practice in the observed overall reduction of antibiotic use in the Danish primary healthcare sector.

## 4. Materials and Methods

The study was conducted as a register-based study, including antibiotic prescriptions issued by GPs in the North Denmark Region between 1 February 2019 to 31 January 2021. Data was extracted from the North Denmark Region prescription database and pseudonymised before being handed over. Antibiotic agents were classified according to the Anatomical Therapeutic Chemical (ATC) system, and both J01 (antibacterial agents for systemic use) and P01AB01 (metronidazole) were included. Information about each prescription (issue date, type of antibiotic), patient (age, gender), and date of any type of consultation in general practice (face-to-face, video, telephone, electronic or home visit) was obtained. The consultation type related to each of the antibiotic prescriptions was found in relation to the antibiotic prescription date by noting any consultation performed within three days (+/−) of the issue. Since prescriptions were noted for patients based on their Central Person Registration number (CPR-number), antibiotics for foreigners were not included in the study. Information about whether the prescription was redeemed at a pharmacy was not available.

To describe the prescribing patterns, data were divided into two comparable periods; a pre-pandemic period running from 1 February 2019 to 31 January 2020 and a pandemic period accounting for 1 February 2020 to 31 January 2021 [31]. To determine the consumption of the various types of antibiotics, the prescriptions were divided into nine antibiotic groups based on the ATC system: penicillins with extended -spectrum (J01CA), beta-lactamase sensitive penicillins (J01CE), beta-lactamase resistant penicillins (J01CF), combinations of penicillins (J01CR), sulphonamides and trimethoprim (J01E), macrolides and lincosamides (J01F), nitrofurantoin (J01XE), metronidazole (P01AB01) and other antibiotics including tetracyclines (J01AA), cephalosporins (J01D), fluoroquinolones (J01MA), polymyxins (J01XB) and methenamines (J01XX) [32].

The number of prescriptions was calculated on a monthly basis, and the difference was noted as percentages. Data were normally distributed, and a paired t-test was conducted to decide if the variation was statistically significant for the two periods. Changes in the distribution of prescriptions within the various groups of antibiotics were determined by using a paired t-test. Missing data were handled by pairwise deletion, as it was not possible to identify a consultation type in relation to all antibiotic prescriptions. Data were analysed using the program IBM SPSS Software version 27.0 [33].

## 5. Conclusions

In conclusion, this study found that the COVID-19 pandemic has had profound impact on antibiotic prescribing patterns in general practices located in the North Denmark Region. This finding indicates that COVID-19 prevention initiatives can probably be of importance moving forward in the fight against increasing problems with antimicrobial resistance. No evidence of an increase in observed invasive infections during the COVID-19 pandemic has been reported, implicating a safe reduction in antibiotic prescribing.

## Figures and Tables

**Figure 1 antibiotics-11-01615-f001:**
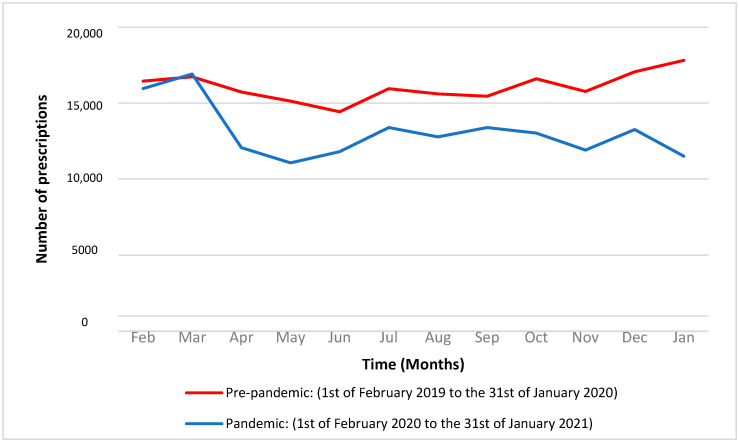
Antibiotic prescribing rates each month before and during the COVID-19 pandemic.

**Figure 2 antibiotics-11-01615-f002:**
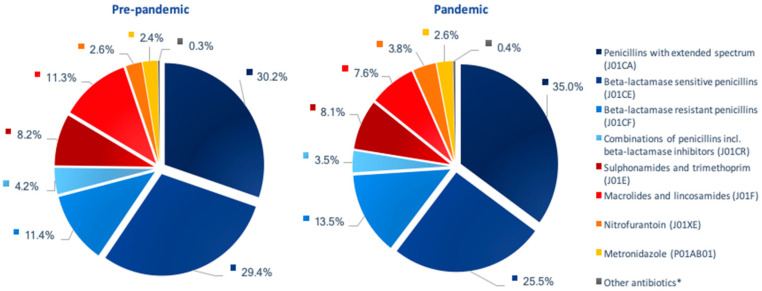
Prescribing of antibiotic agents based on ATC-groups **. * Other antibiotics include tetracyclines (J01AA), cephalosporins (J01D), fluoroquinolones (J01MA), polymyxins (J01XB) and methenamines (J01XX). ** Both piecharts contain identical groups and are directly relatable.

**Figure 3 antibiotics-11-01615-f003:**
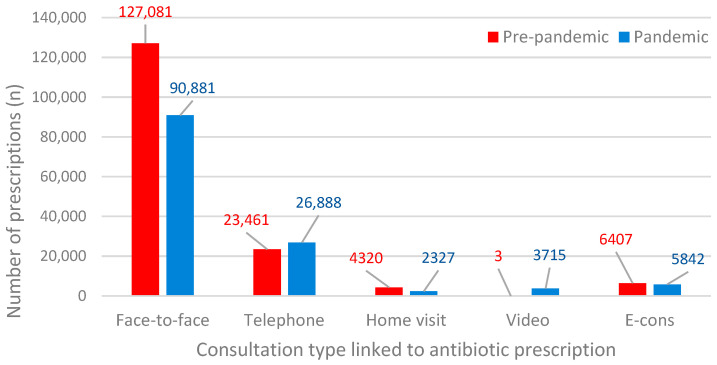
Antibiotic prescriptions are linked to the various types of consultations.

**Table 1 antibiotics-11-01615-t001:** The number of antibiotic prescriptions issued on a monthly basis.

Month	Pre-Pandemic (n) (N = 192,702)	Pandemic (n) (N = 157,065)	Difference (n) *	Difference (%) **
	2019 to 2020	2020 to 2021		
February	16,444	15,951	−493	−3.0
March	16,732	16,918	186	1.1
April	15,727	12,067	−3660	−23.3
May	15,127	11,068	−4059	−26.9
June	14,429	11,804	−2625	−18.2
July	15,946	13,393	−2553	−16.0
August	15,605	12,775	−2830	−18.1
September	15,452	13,389	−2063	−13.4
October	16,599	13,025	−3574	−21.5
November	15,768	11,906	−3862	−24.5
December	17,058	13,261	−3797	−22.3
January	17,815	11,508	−6307	−35.4
Total	192,702	157,065	−35,637	−18.5

* Difference in the number of prescriptions. ** Difference in percentages between the two periods for each month.

## Data Availability

The data presented in this study are not publicly available due to privacy restrictions.

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
