# Peer review of "Changes in Antibiotic Prescribing Patterns in Danish General Practice during the COVID-19 Pandemic: A Register-Based Study"

_antibiotics, 2022, doi:10.3390/antibiotics11111615_

Round 1

Reviewer 1 Report

minor comments 

Title of the study: indicate type of study ….. : retrospective cohort study, RCT, etc

Abstract: please make it structured 

Author Response

Reviewer 1 report:

Title of the study: indicate type of study:
Answer: Thank you very much for this very relevant comment. We have added information about the study design to the title; “a register-based study”.

Abstract: please make it structured:
Answer: Thanks. The abstract has been reorganized to follow the structure suggested in the author instruction of “Antibiotics”; background, aim, methods, results, conclusion. 

Reviewer 2 Report

1) In which year, DANMAP system established and why, discuss in introduction.

2) If antibiotic use decrease, than what is statement of problem. Discuss more clarify.

3) A total reduction of 35,637 antibiotic prescriptions was observed for the pandemic period compared to the pre-pandemic period (p-value <0.0001). This finding corresponds to an overall decrease of 18.5%. Discuss the factors why use of antibiotics is reduced?

4) Discuss more elaborately of results. Its insufficient for peer review. 

5) In which type of infections, the antibiotics use was reduced?

6) Which type of Specialty have impact of COVID.

7) Discuss the reason from published literature, why reduction of antibiotics use.

8) Design of study is missing. Which type of published data was reviewed. 

Author Response

Reviewer 2 report:

Extensive editing of English language and style required

Answer: The manuscript has been re-read thoroughly and adjustments have been made to accommodate this comment.

In which year, DANMAP system established and why, discuss in introduction.

Answer: We agree that this is relevant and have now provided a more thoroughly description of DANMAP in the introduction.

If antibiotic use decrease, than what is statement of problem. Discuss more clarify.
Answer: Thank you for pointing this out. One can argue that the problem formed in advance is the concern expressed by WHO, however if the antibiotic consumption has decreased, we find a potential lead towards handling antibiotic overprescribing in the future. We have adjusted the formulation in the introduction section.

A total reduction of 35,637 antibiotic prescriptions was observed for the pandemic period compared to the pre-pandemic period (p-value <0.0001). This finding corresponds to an overall decrease of 18.5%. Discuss the factors why use of antibiotics is reduced?

Answer: Thanks. This finding is discussed in the “Discussion” section under “Findings in relation to other studies”. Also, the author team has added an additional sentence in “Principal findings” regarding respiratory tract infections.

Discuss more elaborately of results. Its insufficient for peer review. 
Answer: Thanks. The results have been discussed thoroughly in the “discussion” section.  

In which type of infections, the antibiotics use was reduced?
Answer: Thank you very much for pointing this out. We have commented type of infections in the section “Principal findings”.

Which type of Specialty have impact of COVID.

Answer: This study focuses solely on the antibiotic use in Danish general practices, consequently the specialty of relevance for this project is general medicine. However, we have noted under ”Implications”, how other medical specialties in the Danish primary health care sector could be of relevance to investigate further to clarify their consumption compared to general practice.

Discuss the reason from published literature, why reduction of antibiotics use.
Answer: Certainly, this is an important topic to discuss. We have now elaborated on possible reasons for reduced antibiotic use during the COVID-19 pandemic – based on recently published literature.  

Design of study is missing. Which type of published data was reviewed. 
Answer: Thanks for this relevant comment. The author team has added information about the study design to the title; “register-base study”. Data used are not public, but stored at a secure server at Aalborg University. Data was obtained from the North Denmark Region – and pseudonymized before handed over to researchers.

Reviewer 3 Report

No format of research presentation followed. Conclusion of the research was not clearly described. The significance of the Conclusion is not discussed.

Author Response

Reviewer 3 report:

No format of research presentation followed.
Answer: Thanks. The author team has followed the author instruction for “Antibiotics” for “Research manuscripts” – and used the template according to this type of manuscript.

Conclusion of the research was not clearly described. The significance of the Conclusion is not discussed.

Answer: Thank you for pointing this out. A subsection with a conclusion has been added to the manuscript.

Round 2

Reviewer 2 Report

Paper seems good now.

Reviewer 3 Report

Thanks for considering the revision. I am satisfied with the present model.